# Enhancing Clinical Applications by Evaluation of Sensitivity and Specificity in Whole Exome Sequencing

**DOI:** 10.3390/ijms252413250

**Published:** 2024-12-10

**Authors:** Youngbeen Moon, Chung Hwan Hong, Young-Ho Kim, Jong-Kwang Kim, Seo-Hyeon Ye, Eun-Kyung Kang, Hye Won Choi, Hyeri Cho, Hana Choi, Dong-eun Lee, Yongdoo Choi, Tae-Min Kim, Seong Gu Heo, Namshik Han, Kyeong-Man Hong

**Affiliations:** 1Bioinformatics Analysis Team, Research Core Center, Research Institute, National Cancer Center, Goyang 10408, Gyeonggi-do, Republic of Korea; ybeen930@ncc.re.kr (Y.M.); jk@ncc.re.kr (J.-K.K.); 2Cancer Molecular Biology Branch, Division of Cancer Biology, Research Institute, National Cancer Center, Goyang 10408, Gyeonggi-do, Republic of Korea; chhong@ncc.re.kr (C.H.H.); ysh3958@gmail.com (S.-H.Y.); eun14@ncc.re.kr (E.-K.K.); iconpremier@ncc.re.kr (H.W.C.); 3Diagnostic and Therapeutics Technology Branch, Division of Technology Convergence, Research Institute, National Cancer Center, Goyang 10408, Gyeonggi-do, Republic of Korea; youngho5@ncc.re.kr (Y.-H.K.); 76472@ncc.re.kr (H.C.); hnnice87@ncc.re.kr (H.C.); 4Biostatistics Collaboration Team, Research Core Center, Research Institute, National Cancer Center, Goyang 10408, Gyeonggi-do, Republic of Korea; dong-eun@ncc.re.kr; 5Division of Technology Convergence, National Cancer Center, 323 Ilsan-ro, Goyang 10408, Gyeonggi-do, Republic of Korea; ydchoi@ncc.re.kr; 6Department of Medical Informatics and Cancer Research Institute, College of Medicine, The Catholic University of Korea, Seoul 06591, Gyeonggi-do, Republic of Korea; tmkim@catholic.ac.kr; 7Dana Farber Cancer Institute, Boston, MA 02215, USA; luke.hur@gmail.com; 8The Broad Institute of MIT and Harvard, Cambridge, MA 02142, USA; 9Harvard Medical School, Boston, MA 02115, USA; 10Milner Therapeutics Institute, University of Cambridge, Cambridge CB2 0AW, UK; nh417@cam.ac.uk; 11Cambridge Centre for AI in Medicine, Department of Applied Mathematics and Theoretical Physics, University of Cambridge, Cambridge CB3 0WA, UK; 12Cambridge Stem Cell Institute, University of Cambridge, Cambridge CB2 0AW, UK

**Keywords:** WES, false negative error, false positive error, quality control, reference standard, mutation, cancer

## Abstract

The cost-effectiveness of whole exome sequencing (WES) remains controversial due to variant call variability, necessitating sensitivity and specificity evaluation. WES was performed by three companies (AA, BB, and CC) using reference standards composed of DNA from hydatidiform mole and individual blood at various ratios. Sensitivity was assessed by the detection rate of null–homozygote (N–H) alleles at expected variant allelic fractions, while false positive (FP) errors were counted for unexpected alleles. Sensitivity was approximately 20% for in-house results from BB and CC and around 5% for AA. Dynamic Read Analysis for GENomics (DRAGEN) analyses identified 1.34 to 1.71 times more variants, detecting over 96% of in-house variants, with sensitivity for common variants increasing to 5%. In-house FP errors varied significantly among companies (up to 13.97 times), while DRAGEN minimized this variation. Despite DRAGEN showing higher FP errors for BB and CC, the increased sensitivity highlights the importance of effective bioinformatic conditions. We also assessed the potential effects of target enrichment and proposed optimal cutoff values for the read depth and variant allele fraction in WES. Optimizing bioinformatic analysis based on sensitivity and specificity from reference standards can enhance variant detection and improve the clinical utility of WES.

## 1. Introduction

Next-generation sequencing (NGS) is a transformative technology capable of analyzing millions of DNA variants simultaneously [1,2,3]. It has become an essential diagnostic tool, utilized by over 11,000 laboratories and companies across the USA [1]. Despite its advantages, NGS faces challenges related to sensitivity and specificity. In response to these concerns, the Food and Drug Administration (FDA) has issued guidance aimed at facilitating the development of a regulatory framework for NGS testing [4]. Initiatives such as the Sequencing Quality Control 2 (SEQC2) project have sparked numerous studies focused on enhancing NGS quality control [5,6].

Whole exome sequencing (WES) involves the capture, sequencing, and analysis of all protein-coding exons, which constitute less than two percent of the human genome. WES is increasingly utilized in clinical settings and is expected to become the standard of care for various medical conditions in the near future [7,8,9,10,11]. WES has been suggested as an accurate and sensitive method for identifying actionable mutations in cancer [12,13,14,15].

Nevertheless, there is a strong demand among technology users for established standards and guidelines. Several studies have indicated that while WES is associated with lower diagnostic costs, whole genome sequencing (WGS) is deemed the optimal genomic test for maximizing diagnosis in Mendelian disorders [16,17,18,19]. Furthermore, a recent analysis found that WGS, when used as a first-tier analysis, is more cost-effective than employing WES as a first-tier option or in various combinations with WGS [20], which raises questions about the first-tier utility of WES. Compared to WES, WGS offers advantages in the uniformity of read coverage and a more balanced allele ratio [21], underscoring potential issues with the consistency of variant calls in WES. Together, these studies highlight the need for a more thorough evaluation of the sensitivity and specificity of WES results to enhance their clinical implications.

We established a method to evaluate the quality of NGS results and found significant variations in the sensitivity and specificity of targeted NGS results among several NGS-certified service providers [22]. This work is currently under review, but the complete contents are available on bioRxiv (DOI: 10.1101/2024.07.22.603478). In the method, reference-standard samples composed of mixtures of homozygous hydatidiform mole (H. mole) DNA and blood genomic DNA in varying ratios were employed. H. mole, a type of abnormal pregnancy characterized by the proliferation of placental tissue, presents unique genetic characteristics [23,24]. Complete H. mole results from the fertilization of an egg that lacks genetic material by a sperm, leading to a mass of abnormal tissue resembling a cluster of grapes. This type contains only homozygous alleles from the paternal sperm cell, and allowed for the identification of more informative alleles, particularly null and homozygous variant pairs from both H. mole and blood DNAs. These alleles were then employed to estimate the sensitivity of targeted NGS based on the detection rate of diluted informative variants in the reference standards. In the estimation of false positive (FP) errors, FP error alleles are defined as bases present in the mixed reference-standard DNAs but absent in both H. mole and blood genomic DNAs. Three types of allelic pairs in which FP errors occur were defined in the study [22]. Building on this, the present study uses the same reference-standard samples to assess the sensitivity and specificity of WES results from three service providers, certified by either the College of American Pathologists (CAP) or the Ministry of Food and Drug Safety (Korean FDA). It is worth noting that the company names differ from those mentioned in the previous report [22].

## 2. Results

### 2.1. Evaluation of In-House WES Sensitivity

To assess the sensitivity of whole exome sequencing (WES) provided by three service providers, we prepared reference-standard DNAs, including DNA1 (hydatidiform mole DNA with only homozygous alleles) and DNA2 (blood DNA from an individual), as previously described [22]. As illustrated in Figure 1A, there are six N–H pairs, consisting of three null–homozygous pairs (N–Ho, where DNA1 is null and DNA2 contains a homozygous variant) and three homozygous and null pairs (Ho–N, where DNA1 has homozygous variants and DNA2 is null). In this context, null alleles refer to alleles that consist of the reference base at a given site, whereas homozygous alleles consist entirely of the variant base at that site. In reference-standard samples, the variant allelic fraction (VAF) of N–Ho and Ho–N pairs is inversely related based on specific DNA1 to DNA2 ratios. For example, the CH10 reference-standard DNA in Figure 1A has a VAF of 0.9 for the N–Ho pairs and a VAF of 0.1 for the Ho–N pairs. Pairs with a VAF of 0.9 in these reference-standard DNAs are designated as V90, while those with a VAF of 0.1 are referred to as V10. Consequently, the expected VAFs (eVAFs) for the V5 and V95 alleles in reference standards are 0.05 and 0.95, respectively. The detection rate of the diluted variants is assessed relative to the total N–H pairs. In the example shown in Figure 1A, two N–Ho and one Ho–N pair (N–H pair) variant in the reference-standard DNA at an eVAF of 0.05 were detected, resulting in a detection rate of 50% for the V5 alleles.

WES experiments and bioinformatics analyses were carried out by three companies (AA, BB, and CC) utilizing reference-standard DNAs. From the variant calls in the in-house WES results, N–H pairs with a read depth of at least 10 (DP10; read depths for both DNA1 and DNA2 were ≥10 across chromosomes 1 to 22) were selected. The VAFs in these mixed DNA samples were plotted by chromosomal position. In Figure 1B, the VAF values for the alleles with eVAFs equal to or less than 0.5 were plotted. Therefore, variants with a VAF of approximately 1.0 (highlighted by the blue rectangle in Figure 1B) are likely non-specific or false-positive variants. To eliminate these non-specific variants and better assess the sensitivity of WES, a trimming procedure, as described in the Materials and Methods section, was applied to all WES data in the present study, excluding Figure 1. The variants eliminated through trimming, detailed in Appendix A, did not demonstrate a linear relationship between the observed VAF and the eVAF, indicating the effectiveness of the trimming procedure in refining our analysis.

After trimming the N–H pairs, the VAF values from in-house AA, BB, and CC calls were plotted in Figure 1C, Figure 1D, and Figure 1E, respectively. The plot from company AA revealed a significant number of variants with VAFs less than 0.05 (Figure 1C), while there were few variants with VAFs below 0.05 in the results from companies BB and CC (Figure 1D,E), suggesting that variants with low-level VAFs were either not effectively captured by the target enrichment procedure or were eliminated during the in-house analyses for companies BB or CC.

### 2.2. Differences in Sensitivity of In-House WES Results

In estimating the detection rate of diluted variants from the reference standards across various ratios, the lowest variant allelic fraction (VAF) at which 95% of the diluted informative alleles were detected (sensitivity) was found to be 4.82% for company AA and 19.78 and 19.75% for companies BB and CC, respectively (Figure 2A–C). This suggests that the WES results from company AA exhibited better sensitivity compared to those from companies BB and CC. It is noteworthy that company AA utilized a different target enrichment method, specifically Twist, whereas companies BB and CC employed the other target enrichment methods. This variation may contribute to the differences in sensitivity among the three companies.

The total number of detected N–H variants was highest in company BB and lowest in company CC (Figure 2D). The number of detected variants decreased significantly as the read depth cutoffs increased from 10 (DP10) to 50 (DP50) for companies BB and CC, a trend not observed for company AA (Figure 2D). This suggests that read depth was not a significant cutoff in the in-house methods for companies BB and CC, while company AA did employ read depth as an important criterion. When a read depth cutoff of 0 was applied to the results, the sensitivity was comparable to that obtained with a cutoff of 10. Consequently, since a read depth cutoff of 10 has already been used to select N–H variants from DNA1 and DNA2, the data obtained at this cutoff are not presented.

When the various VAF cutoffs from 0 to 0.05 were applied on the N–H variants for sensitivity measurement, the sensitivity of the results from companies BB and CC was essentially the same, but various VAF cutoffs severely affected the results from company AA (Figure 2E). This suggests that VAF was not a significant cutoff in the in-house methods for companies BB and CC, while company AA did employ VAF as an important criterion. Most identified N–H pair variants with the read depth cutoff of 10 from the in-house analyses were common among all three companies (Figure 2F).

### 2.3. Evaluation of WES Sensitivity Using Dynamic Read Analysis for GENomics (DRAGEN) Bioinformatic Analysis

The N–H variants from the WES raw data provided by the three companies were analyzed using the DRAGEN system, resulting in sensitivities that differed from those obtained through in-house analyses. At a read depth of 10, the sensitivity was 10.64% for company AA and around 17.48% and 14.93% for companies BB and CC, respectively (Figure 3A–C). However, the detection rates for V5 alleles (variants with a 5% VAF) exceeded 65% for all three companies, while the detection rates for V10 alleles (variants with a 10% VAF) were well over 90% (Figure 3A–C), indicating higher sensitivity with DRAGEN analysis. Moreover, the number of variants identified by the DRAGEN system significantly surpassed that of the in-house methods: 1.713 times for company AA, 1.339 for company BB, and 1.429 for company CC at a read depth cutoff of 10 (Figure 3D). Consistent with this, the number of variants specific to company AA is notably higher, as illustrated in a Venn diagram (Figure 3E), a trend that was not evident in the in-house results. These suggest that DRAGEN analysis enhances both the detection rate and the overall number of identified variants across WES data providers.

For the N–H variants analyzed using the DRAGEN system, increasing the read depth cutoffs from 10 (DP10) to 50 (DP50) slightly increased sensitivity, as shown in the left column of Figure 3A–C. However, this increase in read depth resulted in a significant decrease in the total number of detected DRAGEN-analyzed N–H variants across all three companies (Figure 3D). This suggests that raising the read depth cutoff can lead to a dramatic reduction in the number of identified variants, while having only a limited effect on sensitivity.

Increasing the VAF cutoffs in DRAGEN-analyzed results for three companies decreases the detection of V5 alleles dramatically, as shown in the right column of Figure 3A–C. Relative to read depth, however, VAF cutoff values up to 0.01 in DRAGEN-analyzed results for all three companies did not affect the sensitivity of the results from all three companies (Figure 3A–C).

For most N–H pair variants from companies AA, BB, and CC, the observed VAFs were well correlated with the eVAFs for V5 to V50 alleles, as shown in Appendix A. This indicates that the majority of variants are true positives, demonstrating a linear correlation between observed VAF and eVAF. Additionally, low-level VAFs (less than 0.05) were identified with DRAGEN analysis in the results from companies BB and CC (Appendix A), which contrasts with the in-house BB and CC results that identified only variants with higher VAFs exceeding 0.05 (Figure 1D,E). These findings suggest that the primary reason for the low sensitivity of in-house results for companies BB and CC was not the capture procedure, but rather the bioinformatic analysis conditions (high VAF cutoff values) used in-house.

### 2.4. Sensitivity Comparisons Among Diverse Categories Based on DRAGEN and In-House Analyses

Next, we analyzed the N–H variants common to both DRAGEN and in-house analyses (C variants), as well as the variants specific to DRAGEN (Dra-Spe variants) or specific to in-house analyses (in-house-Spe variants). C variants identified in DRAGEN analysis are referred to as Dra-C variants, while C variants from in-house analyses are designated as in-house-C variants (AA-C, BB-C, and CC-C). The number of C variants constituted the largest proportion, but there were also significant fractions of Dra-Spe variants for all three companies (Figure 4A). In contrast, the number of in-house specific variants was quite limited, with totals of 155, 250, and 66 for companies AA, BB, and CC, respectively, at a read depth of 10 (Figure 4A). These results indicate that DRAGEN identified most of the N–H variants detected by in-house methods (97.5% for company AA, 96.2% for company BB, and 98.8% for company CC) and effectively identified more variants, primarily Dra-Spe, compared to the in-house analyses.

With the increase in read depth cutoffs from 10 to 50 (DP10 to DP50), the detection rate decreased rather than increased for all companies (Figure 4C,E). This effect is attributed to the loss of low-VAF variants due to the higher read depth cutoff, which is especially evident in the Dra-Spe variants. This trend is also evident in Figure 4A: as the read depth cutoffs increase from 10 to 50, the number of identified Dra-Spe variants decreases dramatically, whereas the number of Dra-C variants remains stable. The read depth of Dra-Spe variants is significantly lower than that of Dra-C variants across all three companies (Figure 4B, *p* < 0.0001 by Mann–Whitney test), suggesting that the dramatic decrease in detection rate of Dra-Spe variants with increased read depth cutoffs is related to their lower read depth.

The sensitivity among the various categories of DRAGEN-analyzed N–H variants revealed that Dra-C variants exhibited higher sensitivity than Dra-Spe variants (Figure 4C,E). For instance, the sensitivity of the DRAGEN-analyzed results from company AA, with a read depth cutoff of 10, was 4.92% for Dra-C variants and 17.86% for Dra-Spe variants (Figure 4C). Therefore, the sensitivity decrease from 5% to 10% with the DRAGEN method compared to the in-house AA method (Figure 3A) is related to the additional Dra-Spe variants, which had significantly lower read depths (Figure 4B). In the DRAGEN-analyzed results from companies BB and CC, the sensitivity at a read depth cutoff of 10 was about 9% for Dra-C variants (9.04% for BB and 9.39% for CC, Figure 4D,E), while it reached 32.59% and 32.43% for Dra-Spe variants from companies BB and CC, respectively, suggesting that the overall sensitivity increase for the DRAGEN-analyzed BB and CC results, compared to in-house results, (Figure 3A–C), is related to the high sensitivity in Dra-C variants. These results indicate that Dra-Spe variants mostly with lower read depths may be related to non-uniform read coverage in WES reported in previous studies [17,20,21].

The changes in sensitivity across various categories with increasing VAF cutoffs were similar to those observed with increasing read depth cutoffs (Appendix A). Specifically: (1) Sensitivity decreased as VAF cutoff values increased; (2) Detection rates for in-house BB and CC were significantly lower than those from company AA, regardless of the VAF cutoffs; (3) The sensitivity of Dra-Spe or in-house-Spe variants was notably lower than that of Dra-C variants, regardless of the VAF cutoffs. Additionally, the results with VAF cutoffs of 0 and 0.01 were essentially the same, indicating that a VAF cutoff of 0.01 does not significantly impact the sensitivity of the DRAGEN-analyzed results.

In the VAF plots for N–H variants according to eVAFs (V5 to V95) in the reference standards, the variations in VAF for in-house specific variants were greater than those for DRAGEN-specific variants (*p* = 0.0002 for the results from all companies according to the Wilcoxon matched-pairs test on the difference between Q1 and Q3, Figure 5A,B). Additionally, the VAF variations in DRAGEN-specific variants were significantly greater than those for Dra-C and in-house-C variants (*p* < 0.0001 for the results from all companies according to the Wilcoxon matched-pairs test on the difference between Q1 and Q3, Figure 5C,D). In contrast, the differences in VAF variation between Dra-C and in-house-C variants were not significant (*p* = 0.0559). An important reason for the higher variability in the in-house-specific and DRAGEN-specific variants, compared to common variants (Dra-C or in-house variants), may be their significantly lower read depths, as shown in Figure 4B. Given that the number of in-house-specific variants is quite limited (less than 4%, Figure 4A), the number of DRAGEN-specific variants becomes the determining factor for achieving uniform variant calls in whole exome sequencing (WES).

### 2.5. Superior WES Sensitivity and Greater Variant Call Rates Observed in Company AA

It is evident that the sensitivity of in-house AA is superior to that of in-house BB or CC (Figure 2A–C), as more V5 and V10 alleles were effectively identified in the in-house AA results. However, when analyzed using the DRAGEN system, the sensitivity differences among the results from the three companies decreased (Figure 3A–C). Notably, the sensitivity difference for Dra-C variants from three companies was not apparent (Figure 4C–E). This suggests that the higher WES sensitivity of company AA compared to companies BB and CC is largely attributable to its in-house analytic conditions.

Nevertheless, the total number of DRAGEN AA calls was significantly greater than those from DRAGEN BB and CC (Figure 6A), and the difference in sensitivity with the DRAGEN system still persisted (Figure 3A–C). This suggests that other factors, including greater sequencing depth and more efficient target enrichment employed by company AA, may contribute to small differences in sensitivity.

We found a substantial difference in the number of DRAGEN-specific N–H variants from three companies. The total N–H variant calls in DRAGEN-analyzed AA variants were 1.20 and 1.31 times greater than those from companies BB and CC, respectively (Figure 6A). In contrast, a similar number of Dra-C variants were identified across the three companies: relative to the variant count from company AA, the numbers from companies BB and CC were 1.05 and 0.93, respectively. However, the DRAGEN-specific variants from company AA were 1.84 and 1.83 times greater than those from companies BB and CC, respectively (Figure 6A). These findings suggest that the differences in the number of variants identified by the DRAGEN method among the three companies are primarily related to the quantity of DRAGEN-specific variants.

The median read depth of the AA Dra-C variants was 2.05 and 2.50 times higher than that of the BB and CC Dra-C variants, respectively (Figure 4B). Consistent with this, the median sequencing depth of WES from company AA was 1.96 and 1.74 times higher than that of BB and CC companies, respectively (Appendix A). Therefore, the increased number of Dra-Spe variants from company AA may be attributed to the higher sequencing depth. However, this difference may be influenced by another factor: the Twist exome enrichment system, which is utilized exclusively by company AA.

The higher total number of N–H variants identified by company AA compared to the other two companies, even with a read depth cutoff increased to 50 (Figure 6B), suggests that the Twist target enrichment employed by company AA may contribute to its higher sensitivity and greater number of variant calls, though this effect is less pronounced than that achieved through effective bioinformatic analysis.

To further investigate the contribution of Twist exome enrichment for greater call numbers in company AA, we compared the variances in VAFs and eVAFs for Dra-C and Dra-Spe variants from the three companies using a read depth cutoff of 10. The VAF variations for AA Dra-Spe variants showed significant differences when compared to those of companies BB and CC (*p* = 0.0156 for AA vs. BB and AA vs. CC, according to the Wilcoxon matched-pairs test on the differences between Q1 and Q3, Figure 6C). In contrast, there was no significant difference between the VAFs of BB and CC Dra-C variants (*p* = 0.5781). Additionally, for AA Dra-C variants, the VAF variations were also significantly different from those of companies BB and CC (*p* = 0.0156 for AA vs. BB and AA vs. CC, according to the Wilcoxon matched-pairs test on the differences between Q1 and Q3, Figure 6D). However, no significant difference was observed between BB and CC Dra-Spe variants (*p* = 0.5781). These results suggest that the higher repeatability or smaller variation in VAF for Dra-C and Dra-Spe variants in the results from company AA is likely due to the Twist exome enrichment method rather than differences in sequencing depth, since increased sequencing depth does not necessarily enhance VAF repeatability.

A previous report [25] noted that Twist target enrichment is advantageous for capturing sequences with higher GC content. However, the GC content of total DRAGEN-analyzed variants among the three companies was similar or rather lower (Figure 6E), indicating that sequences with higher GC contents may not be effectively enriched by the Twist target enrichment system used by company AA. Instead, the GC contents for Dra-Spe variants, which represent a major factor for higher variant detection in the results from company AA, were significantly lower than those for Dra-C variants (*p* < 0.0001 for each three company according to the Mann–Whitney test) in Figure 6E, suggesting that enrichment of the sequences with lower GC rather than higher GC contents may be the major factor in the Twist enrichment system. It is also noteworthy that the GC contents of Dra-Spe variants from company AA were significantly lower than those from company CC (*p* = 0.0214 according to the Mann–Whitney test) in Figure 6E.

In the analysis of read depth distributions for DRAGEN-analyzed variant calls in DNA1 from the three companies, the distributions appeared similar (Figure 6F–H). In the figure, the count of variants at the same read depths (red) and the total read depth count (blue) are shown to illustrate the read depth distribution. The total read depth count was calculated by multiplying the count of variants by their respective read depths. This similarity was also observed in the DNA2 sample results (Appendix A). Furthermore, the analysis of in-house results revealed no apparent differences among the three companies (Appendix A).

### 2.6. Analysis of False Positive Errors in WES Results

By employing reference-standard DNAs, false positive (FP) errors in WES results can be estimated. FP error alleles are defined as bases present in the mixed reference-standard DNAs but absent in both DNA1 and DNA2. FP errors can occur in three types of paired alleles: (1) R–R Pair Alleles: Both DNA1 and DNA2 have the reference base; (2) V–V Pair Alleles: Neither DNA1 nor DNA2 has the reference base; (3) R–V Pair Alleles: DNA1 and DNA2 contain both reference and variant bases. In the example illustrated in Figure 7A, there is one FP error in the R–R pair alleles, two in the V–V pair alleles, and one in the R–V pair alleles.

First, FP errors from R–R pairs were analyzed. In the in-house results from company AA, increasing the VAF cutoff levels (0, 0.01, 0.03, and 0.05) significantly reduced FP errors, while increasing the read depth cutoff (10, 20, 30, and 50) had little effect on the FP rate (Figure 7B). In contrast, for the in-house results from companies BB and CC, increasing the VAF cutoff did not impact FP errors, but increasing the read depth cutoff resulted in a substantial reduction in FP errors (Figure 7D,F). These results may be attributed to two factors: (1) the relatively high read depth cutoff in the in-house AA analysis and (2) the relatively high VAF cutoffs used in the in-house BB and CC analyses.

The median total false positive (FP) errors from R–R pairs in in-house analyses varied greatly, ranging from 4.07 (1844 for AA, and 453 for BB) to 13.97 (1844 for AA and 132 for CC) times (Figure 7B,D,F). However, in the analyses performed with DRAGEN, the differences were smaller, ranging from 1.70 (979 for AA, and 576 for CC) to 2.07 (979 for AA, and 474 for CC) times when using a read depth cutoff value of 10 (Figure 7C,E,G). With higher read depth and VAF cutoff values, the decrease in differences from the DRAGEN analysis was maintained. The median total FP error from R–R pairs was reduced from 1844 for in-house AA results to 979 for DRAGEN AA results (0.53 times). Although the total FP errors from R-R pairs were 1.05 and 4.36 times higher in the DRAGEN-analyzed results for companies BB and CC, respectively, than in-house results, the sensitivity of the in-house results was only 20%, likely due to the use of an extreme VAF cutoff condition (over 0.05), indicating that DRAGEN-analyzed results with a read depth cutoff of 10 showed dramatically higher sensitivity with less damage in specificity.

In the variant calls from DRAGEN, both VAF and read depth cutoffs significantly affected FP errors (Figure 7C,E,G), indicating that both cutoffs were not applied in the DRAGEN analysis. With the DRAGEN analysis, over 30,000 FP errors were observed without a read depth cutoff, suggesting that using a read depth cutoff is essential for reducing FP errors (Figure 7C,E,G). Furthermore, a read depth cutoff above 10 did not lead to a substantial reduction in FP errors. The other FP errors from V–V and R–V pairs were relatively small in the WES data (Appendix A). Therefore, these were not included in the analysis.

## 3. Discussion

In the present study, we observed substantial variability in the sensitivity (approximately four-fold) and specificity (ranging from 4.07 to 13.97-fold) of whole exome sequencing (WES) results from three service providers certified by either the College of American Pathologists (CAP) or the Ministry of Food and Drug Safety (Korean FDA). The application of the DRAGEN system for bioinformatic analysis, using a read depth cutoff of 10, reduced sensitivity differences among service providers and narrowed the range of false positive (FP) errors (from 13.97 to about 2 times). Additionally, DRAGEN identified 1.3 to 1.7 times more variants than in-house bioinformatic methods at the same read depth cutoff. Therefore, our findings suggest that effective bioinformatic analysis for WES, employing cutoff values based on sensitivity and specificity from reference standards, can enhance sensitivity and increase variant detection without a significant loss of specificity.

Although concerns about the sensitivity of WES have been raised [26], it has rarely been thoroughly evaluated [27]. Initially considered straightforward, NGS analysis has proven prone to errors [28], contradicting claims that NGS errors are negligible [29,30,31]. In WES studies using two cell line databases, namely, the Genomics of Drug Sensitivity in Cancer (GDSC) and the Cancer Cell Line Encyclopedia (CCLE), inconsistent mutation calls were reported to reach as high as 43% [32], indicating that WES results may contain significant errors. One study suggested that discrepancies in mutations between the two consortia may arise from genetic evolution during cell line maintenance in separate laboratories [33], assuming that NGS errors are negligible. However, by performing targeted next-generation sequencing (T-NGS) on 151 genes in 35 common cell lines from GDSC and CCLE, we previously reported a high rate (40–45%) of FN errors in the databases, contributing to the high rate of inconsistent mutation calls [28]. In another study [22], we demonstrated significant variations in the sensitivity and specificity of targeted NGS, related to FN and FP errors. In the present study, we observed significant variations in the sensitivity and specificity of WES, particularly noting high rates of FN errors in the in-house results from companies BB and CC. Consequently, it seems that errors in NGS are inevitable, making it a major concern to identify effective strategies for controlling these error rates. Thus, the evaluation method used in this study provides a strong foundation for developing strategies that enhance the detection of low AF mutations in cancer tissues, as well as mutations in mosaic rare diseases.

The present study examines sensitivity issues related to WES [5,21]. The results indicated that companies BB and CC had lower sensitivities (18–19%) compared to company AA (approximately 5%), suggesting that AA’s method may address these sensitivity issues. When DRAGEN bioinformatic analysis was applied, the sensitivity for company AA decreased (to 10.64%), while it increased for companies BB and CC (15–17%). This suggests that DRAGEN analysis performs moderately compared to the three in-house methods. However, DRAGEN identified 1.34 to 1.71 times more variants than the in-house methods, leading to the classification of variants into categories such as Dra-C and Dra-Spe. Sensitivity for Dra-C variants was 4–5% for company AA and about 9% for companies BB and CC, indicating that DRAGEN analysis is comparable to company AA’s in-house method. For Dra-Spe variants, which comprise 30–70% of Dra-C variants, sensitivity was 14–16% for company AA and 20% for companies BB and CC, which explains the decrease in sensitivity observed in DRAGEN-analyzed results for company AA. Notably, Dra-Spe variants showed a significant decrease in the number of identified variants with an increased cutoff read depth, correlating with relatively lower read depth. Therefore, employing DRAGEN analysis on existing WES databases, such as the CCLE and GDSC [32], could improve the overall sensitivity and consistency of mutation calls. Additionally, improving the target capture of Dra-Spe variants may further increase WES sensitivity and address issues related to non-uniform read coverage [17,20,21].

Although effective bioinformatic analyses are essential [34,35], new technologies, such as efficient target enrichment, have the potential to significantly enhance the quality of WES raw data. Various target enrichment methods have been employed in NGS analyses [36,37,38]. Notably, company AA utilized a different target enrichment system, Twist, which may account for its enhanced sensitivity. Recent reports have highlighted Twist’s superior performance in target enrichment [25]. Therefore, the fourfold higher sensitivity observed in-house at company AA compared to other companies led us to assume that Twist target enrichment improves WES sensitivity. However, DRAGEN analyses of the raw data from all three companies indicated that the majority of the differences in sensitivity stemmed from varying in-house bioinformatic conditions.

In further analyses, a greater number of variants (primarily Dra-Spe variants) were identified in results from company AA, even at an increased read depth cutoff of 50, suggesting that Twist target enrichment contributes to the increased number of variant calls. Additionally, the variances in variant allele frequencies (VAFs) for Dra-C and Dra-Spe variants from company AA were significantly lower than those from other companies, again indicating a possible role for Twist target enrichment in reducing variability in WES variant calls. This mechanism may be linked to the efficient capture of sequences with lower GC content, contrary to suggestions made in a previous report [25]. However, further studies utilizing the same sequencing depth are necessary to clarify the potential effects of efficient target enrichment systems on WES variant calls. Additionally, any technical variations introduced by company AA during the enrichment procedure, which may enhance sensitivity and increase variant calls, should also be controlled for.

Efforts to reduce false positive (FP) errors in WES can negatively impact sensitivity, and vice versa, especially when strict bioinformatic criteria are applied [39]. The primary factors influencing FP errors and sensitivity in bioinformatic analyses are the cutoff values for variant allele frequency (VAF) and read depth [40,41,42,43,44,45], both of which significantly affect sensitivity. In this study, we demonstrated how variations in these cutoff values affect FP errors by analyzing the in-house results from three companies. DRAGEN analyses further highlighted the influence of these cutoff values on FP errors across the WES datasets from all three companies. Our analysis showed that using a read depth cutoff of less than 10 resulted in a substantial number of FP errors, while increasing the cutoff beyond 10 did not significantly reduce FP errors. Additionally, higher read depth cutoffs led to a notable reduction in both sensitivity and the number of detected variants. These findings suggest that the optimal read depth cutoff for DRAGEN analysis is around 10, balancing sensitivity and specificity without causing a significant increase in FP errors.

The total FP errors from the DRAGEN analyses, using a read depth cutoff of 10, were intermediate compared to the three in-house results. However, the sensitivities of the in-house BB and CC results were only 20%, likely due to the use of a VAF cutoff of approximately 0.05 for variant selection. This suggests that higher VAF cutoffs can significantly reduce sensitivity. Additionally, applying a VAF cutoff of up to 0.01 had no impact on sensitivity. These findings indicate that a VAF cutoff of around 0.01 may be optimal for DRAGEN-analyzed results. Therefore, adjusting the DRAGEN results to consider sensitivity and specificity could provide the best reasonable trade-off [22].

Several limitations of this study should be acknowledged: (1) The focus was on base changes rather than deletions or gains; (2) While clinical samples are typically formalin-fixed and paraffin-embedded (FFPE), this study did not analyze WES results from FFPE samples; (3) The study exclusively utilized Illumina systems for next generation sequencing platforms. Further research should investigate performance across different platforms; (4) Technical bioinformatic details from the companies were unavailable, which limits our understanding of the observed differences in sensitivity and specificity.

This study revealed significant variations in sensitivity and specificity among WES results from three certified NGS service providers. While innovations are often prioritized in the clinical implementation of precision medicine models [46], challenges such as the complexity of NGS raw data processing and the absence of standardized protocols may hinder the effective application of WES and lead to incorrect conclusions [47,48]. Therefore, robust quality controls using standardized reference materials are essential for the clinical application of WES.

## 4. Materials and Methods

### 4.1. Preparation of Reference-Standard DNAs

Reference-standard DNAs were prepared as previously described [22]. Two types of DNA were utilized: DNA1, which consists of H. mole DNA containing only homozygous alleles, and DNA2, derived from blood samples of an individual. Blood samples were obtained with informed consent, and the study received approval from the Institutional Review Board (IRB) of the National Cancer Center in Korea. Genomic DNA was extracted from the blood samples using the DNeasy Blood and Tissue Kit (Qiagen, Valencia, CA, USA). Human H. mole DNA (NA07489, Coriell, Camden, NJ, USA) was further purified with QIAamp DNA Mini Kits to eliminate RNA contamination. Mixed samples were prepared by combining DNA1 and DNA2 in various ratios to create a range of variant allele frequencies (VAFs). The following mixtures were created: CH5 (5:95), CH10 (10:90), CH20 (20:80), CH50 (50:50), CH80 (80:20), CH90 (90:10), and CH95 (95:5), with the ratios of DNA1 to DNA2 indicated in parentheses. DNA concentration and purity were assessed using a Nanodrop 8000 UV-vis spectrometer (Thermo Scientific, Waltham, MA, USA) and a Qubit 2.0 Fluorometer (Life Technologies, Grand Island, NY, USA). The reference standards were then sent to service providers for whole exome sequencing (WES).

### 4.2. Whole Exome Sequencing (WES)

WES was conducted by three service providers (AA, BB, and CC) on the reference-standard DNAs, which included unmixed samples of DNA1 and DNA2, as well as their mixtures. The total amount of DNA requested for WES from each company was 500 ng for each sample. Each company utilized proprietary target capture methods and sequencing platforms, all based on Illumina sequencing technology. Before sequencing, the quality and quantity of genomic DNA were reassessed according to each company’s standard protocols. Exome capture, paired-end NGS library construction, and sequencing were carried out following each company’s proprietary methods. Company AA is CAP-accredited, company BB is accredited by the Korean FDA, and company CC is accredited by both CAP and the Korean FDA for NGS services. For exome enrichment, company AA utilized an exome capture kit from Twist Biosciences (South San Francisco, CA, USA), while the other two companies employed different target enrichment methods, though detailed information on their kits was not provided. The captured libraries were sequenced on Illumina NovaSeq 6000 platforms. Quality control measures were implemented at each step of the process, including assessments of DNA fragmentation, library size distribution, and capture efficiency by each company. Total reads and the percentage of bases over Q30 for WES from companies AA, BB, and CC are shown in Appendix A. In our analyses of in-house BAM files from the three companies, duplicate reads were identified in the WES results from company AA.

### 4.3. Variant Calling

Variants were analyzed using two approaches: each company’s proprietary bioinformatics methodologies and the DRAGEN system (version 4.2, Illumina Inc., CA, USA, USA). In the in-house method, the company’s proprietary techniques for general alignment and variant calling were utilized; however, specific details were not disclosed. For the DRAGEN analysis, we utilized the Illumina DRAGEN (Dynamic Read Analysis for GENomics) pipeline with WES FASTQ files. The sequencing data was aligned to the GRCh37 (hg19, GCF_000001405.13) reference genome using the DRAGEN alignment module. Following alignment, the variant calling was performed with default settings to generate variant call format (VCF) files.

The human reference genome hg38 was used for the alignment of sequences to make BAM files in the in-house BB method, and the procedure for conversion from genome hg38 to hg19 was performed by LiftOver [49].

Informative alleles from variant calls generated using in-house or DRAGEN methods, specifically, N–H pairs in chromosomes 1 to 22, were analyzed to assess false negative (FN) errors and the sensitivity of WES. Only variants with a read depth of at least 10 for both DNA1 and DNA2 were included in the analysis.

To remove non-specific or false positive (FP) variants from N–H pairs, we performed separate procedures for N–Ho pairs and Ho–N pairs. The V5 to V95 groups from both procedures were then combined for further analysis.

To trim N–Ho pairs, we undertook the following steps: First, we selected N–Ho pairs based on variant data and VAF from the WES variant call data of DNA1 and DNA2. Variants with a VAF of 0.05 or less in DNA1 were labeled as null, while those with a VAF of 0.95 or higher in DNA2 were labeled as homozygous. Next, we excluded N–Ho pairs with a read depth of fewer than 10 in DNA2, CH5, CH10, and CH20 samples. We also removed pairs with a VAF of 0.7 or higher in CH80, CH90, and CH95, as well as those with a VAF of 0.8 or higher in CH50. In the final results, the trimmed N–Ho pairs were labeled from CH5 to CH95 as V95 to V5 in reverse numeric order.

For Ho–N pairs, we followed a similar procedure to eliminate non-specific or FP variants. We began by selecting Ho–N pairs using variant data and VAF from DNA1 and DNA2. Variants with a VAF of 0.05 or lower in DNA2 were considered null, while those with a VAF of 0.95 or higher in DNA1 were labeled as homozygous. Ho–N pairs with a read depth of fewer than 10 in DNA1, CH95, CH90, and CH80 were excluded from the list. We then removed pairs with a VAF of 0.7 or higher in CH5, CH10, and CH20, as well as those with a VAF of 0.8 or higher in CH50. In the final results, the trimmed Ho–N pairs were labeled from CH5 to CH95 as V5 to V95 in ascending numeric order. The combined total of V5 to V95 from both N–Ho and Ho–N pairs are classified as N–H pair variants in the V5 to V95 groups, with an expected VAF (eVAF) ranging from 0.05 to 0.95.

The detection rate of diluted variants was calculated at various eVAFs. Sensitivity was defined as the lowest percentage of allele fraction at which 95% of the diluted informative alleles were detected. The analysis was conducted using various read depth and VAF cutoffs. For instance, with a read depth cutoff of 10, variants with a read depth of less than 10 were considered null. Similarly, for a VAF cutoff of 0.01, variants with a VAF less than 0.01 were considered null.

### 4.4. Comparison of Variant Calls in Various Categories

Variants were categorized as follows: 1. Dra-C: DRAGEN-analyzed variants that are common to both DRAGEN and in-house identified variants. 2. In-house-C: In-house-analyzed variants common to both DRAGEN and in-house methods, which include three subtypes: AA-C (AA in-house-C variants), BB-C (BB in-house-C variants), and CC-C (CC in-house-C variants). 3. Dra-Spe: DRAGEN-analyzed variants that are not present in the in-house variants. 4. In-house-Spe: In-house-specific variants not found in DRAGEN-analyzed variants, which include three subtypes: AA-Spe (AA in-house-specific variants), BB-Spe (BB in-house-specific variants), and CC-Spe (CC in-house-specific variants).

### 4.5. False Positive (FP) Error Analysis

FP errors were assessed using the reference-standard DNAs, with FP error alleles defined as bases present in the mixed samples but absent in both DNA1 and DNA2. These errors were categorized into three types, as previously reported [22]: R–R pair, V–V pair, and R–V pair alleles. FP error rates were calculated under various conditions based on VAF and read depth cutoffs.

### 4.6. Statistical Analysis and Visualization

Differences in variant numbers, read depths, and VAF distributions were analyzed using appropriate statistical tests. Comparison for read depth and GC contents was tested using the Mann–Whitney test. The variation in VAF values was measured using the interquartile range (IQR), defined as the difference between the first quartile (Q1) and the third quartile (Q3). The IQR was visualized alongside Q1, the median, and Q3. To conduct pairwise comparisons between companies, the nonparametric Wilcoxon matched pairs test was employed, pairing groups with the same eVAF. The null hypothesis posits that there is no difference in the IQR between companies, while the alternative hypothesis suggests that a difference exists.

The sensitivity of WES was defined as the lowest eVAF at which ≥ 95% of N–H variants are detected. To determine this lowest eVAF for calculating sensitivity, Probit regression, a linear regression model that employs a cumulative normal distribution link function, was used to estimate a regression line based on either all or a portion of the observed values.

For data visualization, various graph types were employed. VAFs were plotted by chromosomal positions. Box plots with 10–90 percentile whiskers were used to visualize VAF distributions. Read depth distributions were presented as histograms.

## 5. Conclusions

This study highlights considerable variability in the sensitivity and specificity of whole exome sequencing (WES) results across various certified next-generation sequencing (NGS) service providers. Although advancements in bioinformatics and target enrichment technologies, such as DRAGEN and Twist, demonstrate potential for enhancing WES data quality, inconsistencies remain that could impact clinical applications. This underscores the necessity for robust quality control measures and standardized reference materials to ensure accurate and consistent WES results.

## Figures and Tables

**Figure 1 ijms-25-13250-f001:**
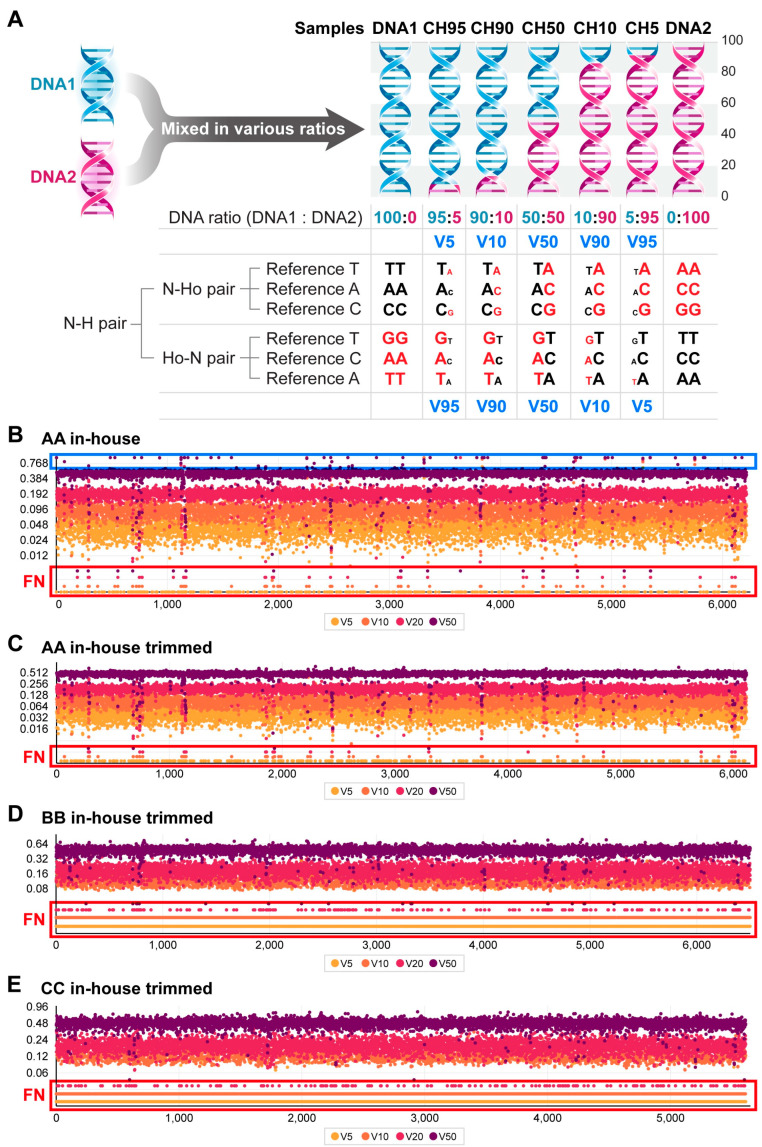
Evaluation of WES sensitivity by estimating false negative (FN) errors in reference standards. (**A**). Sensitivity determination using WES results from reference-standard DNAs prepared by mixing DNA1 (H. mole DNA) and DNA2 (blood DNA) at various ratios. Two types of N–H pair alleles were analyzed: N–Ho pairs and Ho–N pairs. Variant alleles detected are marked in red. Variants with expected variant allele fractions (eVAFs) ranging from 0.05 to 0.95 are labeled as V5 to V95 (in blue). (**B**). Observed VAF and FN errors of in-house AA N–H pairs after removing variants on the X or Y chromosome. A significant number of variant alleles with eVAFs of 50% or less displayed VAFs around 1 (blue box). (**C**). FN errors of in-house AA N–H pairs after trimming. Details for trimming are provided in Materials and Methods. (**D**). FN errors of in-house BB N–H pairs after trimming. (**E**). FN errors of in-house CC N–H pairs after trimming. For panels (**B**–**E**), the FN alleles in the reference standards are highlighted within the red rectangles. The details regarding V5 to V50 are provided in the Materials and Methods section. *X*-axis: alleles aligned by chromosomal position. *Y*-axis: observed VAF.

**Figure 2 ijms-25-13250-f002:**
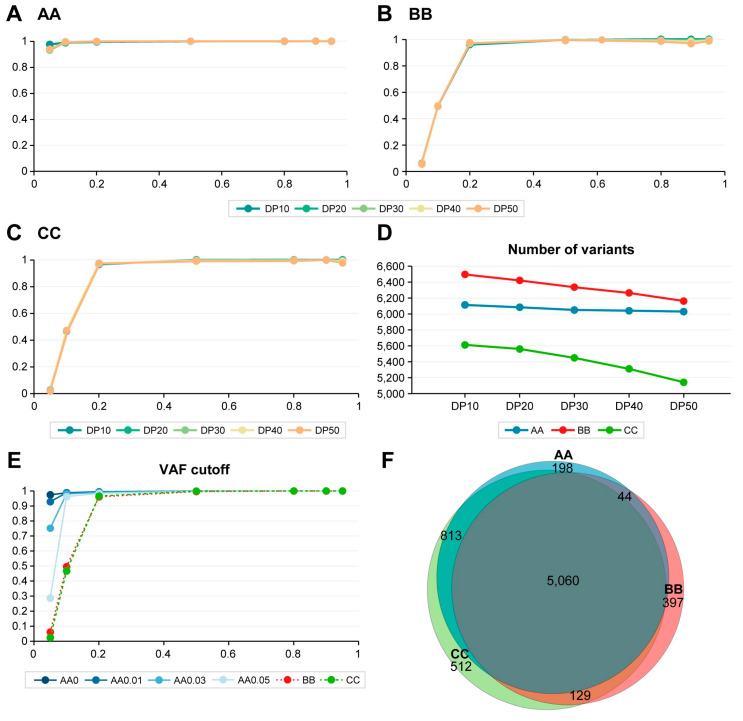
Variant detection rate and the total number of in-house N–H pairs at various eVAFs. A. Variant detection rate of in-house AA results according to read depth cutoffs. B. Variant detection rate of in-house BB results according to read depth cutoffs. C. Variant detection rate of in-house CC results according to read depth cutoffs. In panels (**A**–**C**), the detection rate (*Y*-axis) is plotted against expected variant allele frequencies (eVAFs) (*X*-axis) for various read depth cutoffs ranging from 10 to 50 (DP10 to DP50). (**D**). Number of detected variants based on read depth cutoffs from 10 to 50 (DP10 to DP50) for the three companies. The *X*-axis represents read depth cutoffs, while the *Y*-axis shows the number of detected variants. (**E**). Variant detection rate based on in-house results across VAF cutoffs ranging from 0 to 0.05. The detection rate (*Y*-axis) is plotted against eVAFs (*X*-axis) for different VAF cutoffs, labeled as AA0 to AA0.05. Detection rates for companies BB and CC were consistent across all VAF cutoffs from 0 to 0.05. (**F**). A Venn diagram illustrating in-house variant calls for companies AA, BB, and CC with a read depth cutoff of 10.

**Figure 3 ijms-25-13250-f003:**
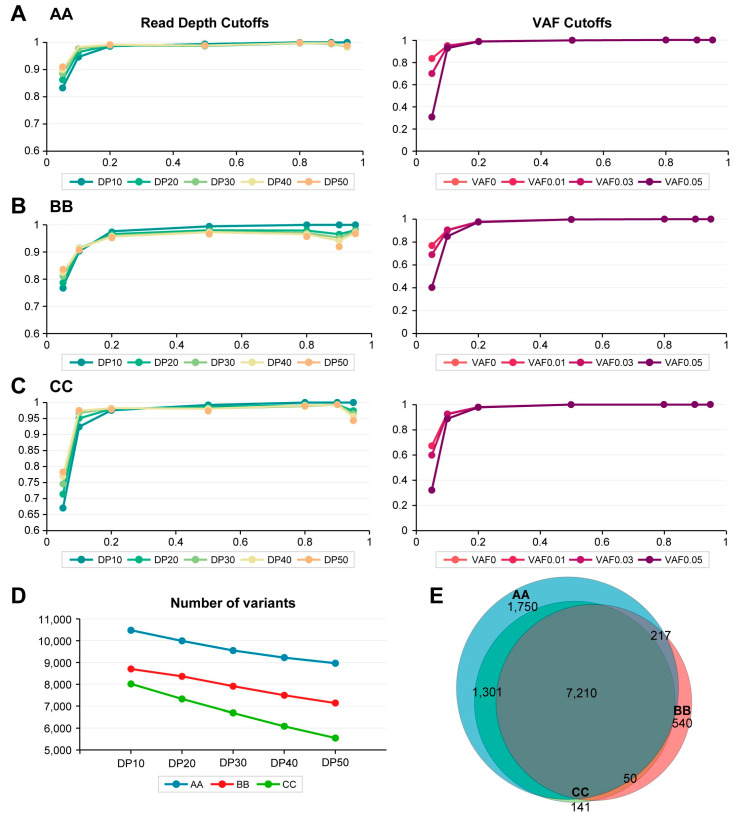
Detection rates of N–H variants analyzed by the DRAGEN system in whole exome sequencing (WES) results from three companies. (**A**). Detection rate of DRAGEN-analyzed variants for company AA. (**B**). Detection rate of DRAGEN-analyzed variants for company BB. (**C**). Detection rate of DRAGEN-analyzed variants for company CC. For panels (**A**–**C**), the detection rate (*Y*-axis) is plotted against eVAF values (*X*-axis). The left column applies read depth cutoffs ranging from 10 to 50 (DP10 to DP50), while the right column employs variant allele frequency (VAF) cutoffs from 0 to 0.05 (VAF0 to VAF0.05). (**D**). Number of variants detected based on read depth cutoffs (DP10 to DP50). *X*-axis: the read depth cutoff. *Y*-axis: the number of detected variants. (**E**). A Venn diagram illustrating the DRAGEN-analyzed WES results from three companies shows that most N–H variants were shared among them.

**Figure 4 ijms-25-13250-f004:**
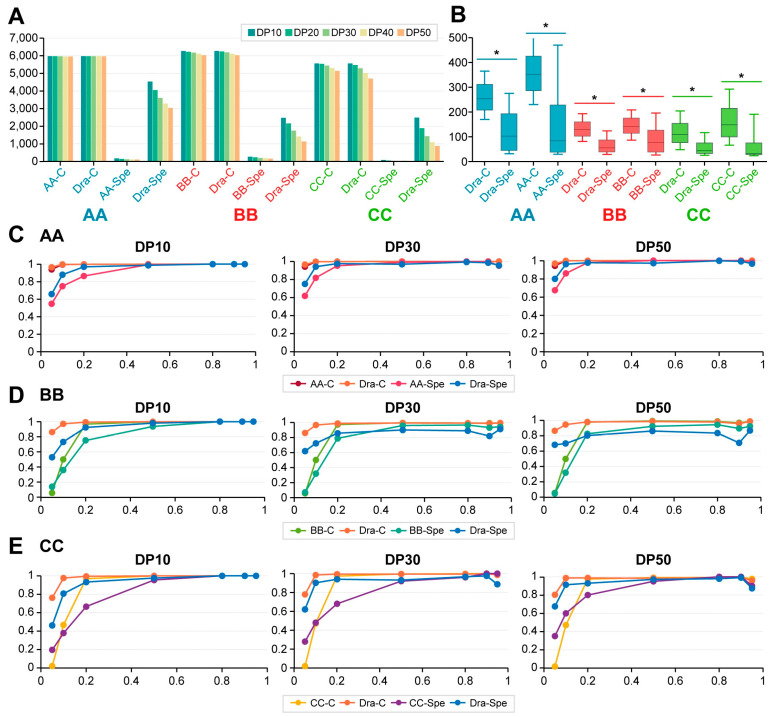
The number and detection rate of variants common to or specific to in-house or DRAGEN analyses. (**A**). The number of N–H variants analyzed by DRAGEN or in-house methods across various categories, indicating commonality or specificity. *Y*-axis: read depth. (**B**). Read depth in combined N–H alleles from DNA1 and DNA2 with a read depth cutoff of 10. *Y*-axis: read depth expressed as a box plot with 10–90 percentile whiskers. * indicates *p* < 0.0001. (**C**). Detection rate of N–H variants in AA results depending on read depth cutoffs from 10 to 50 (DP10 to DP50) across various categories. (**D**). Detection rate of N–H variants in BB results across various categories. (**E**). Detection rate of N–H variants in CC results across various categories. In panels (**C**–**E**), the *X*-axis represents eVAF and the *Y*-axis represents the detection rate of variants. For all panels, Dra-C, AA-C, BB-C, CC-C, Dra-Spe, AA-Spe, BB-Spe, and CC-Spe can be found in the Materials and Methods section.

**Figure 5 ijms-25-13250-f005:**
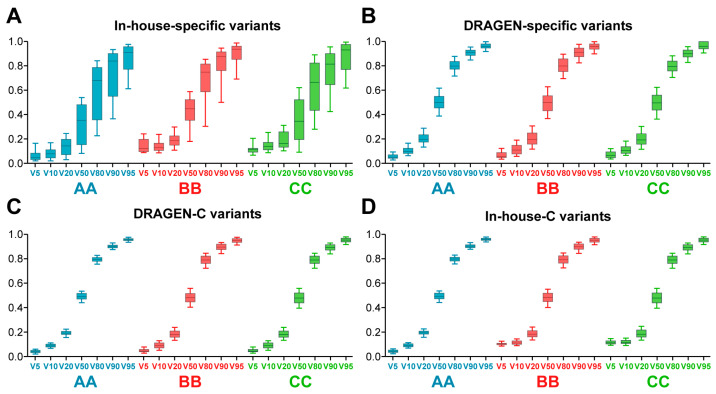
Observed VAF of N–H variants according to expected variant allele frequencies (eVAFs). (**A**). Observed VAF of in-house-specific N–H variants. (**B**). VAF of DRAGEN-specific N–H variants. (**C**). VAF of Dra-C N–H variants. (**D**). VAF of in-house-C N–H variants. The *X*-axis represents eVAF, ranging from V5 to V95, for companies AA, BB, and CC. The *Y*-axis displays VAF as a box plot with 10–90 percentile whiskers.

**Figure 6 ijms-25-13250-f006:**
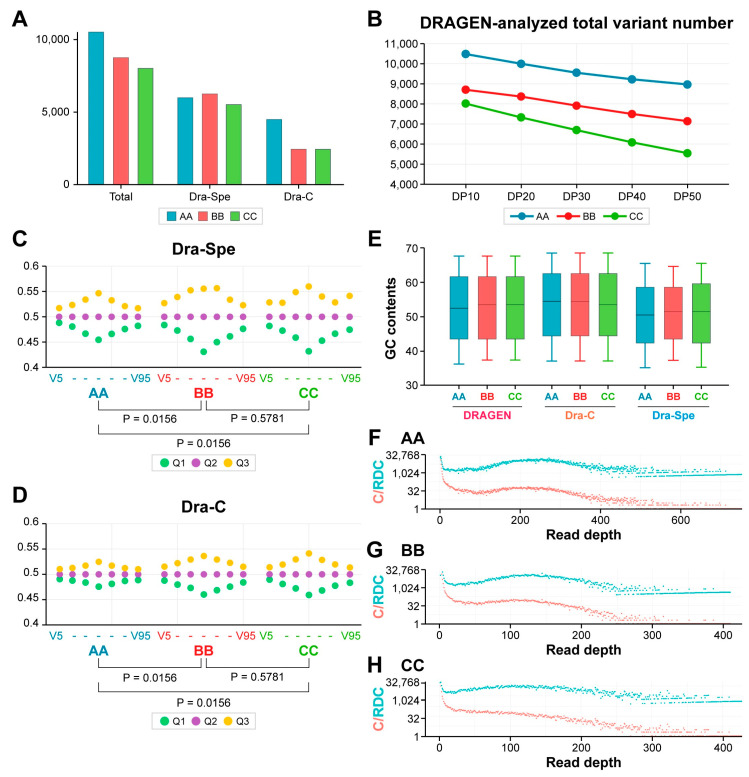
Characteristics of DRAGEN-analyzed N–H variants from the three companies. (**A**). The number of identified N–H variants. The *Y*-axis represents the number of N–H variants. (**B**). The change in the total number of DRAGEN-analyzed N–H variants according to read depth cutoffs. *X*-axis: read depth cutoffs from 10 to 50 (DP10 to DP50). *Y*-axis: the number of identified N–H variants. (**C**). Variation in VAF for Dra-Spe variants. (**D**). Variation in VAF for Dra-C variants. In panels (**C**,**D**), the *Y*-axis represents the adjusted VAF values, with the Q2 value adjusted to 0.5, while the differences are applied also to the Q1 and Q3 values. *X*-axis: variants from V5 to V95. The statistical significance is displayed below the graphs. (**E**). Comparison of GC content for DRAGEN-analyzed total (DRAGEN), Dra-C, and Dra-Spe variants, using a low read depth cutoff of 10. *Y*-axis: GC contents (%) as a box plot with 10–90 percentile whiskers. (**F**). Read depth distributions for DRAGEN-analyzed variants from Company AA. (**G**). Read depth distributions for DRAGEN-analyzed variants from company BB. (**H**). Read depth distributions for DRAGEN-analyzed variants from company CC. For panels (**F**–**H**), the DNA1 sample was analyzed. *X*-axis: the read depth of variants with the same read depth. *Y*-axis: the count of variants with the same read depths (in red) alongside the total read depth count (in blue).

**Figure 7 ijms-25-13250-f007:**
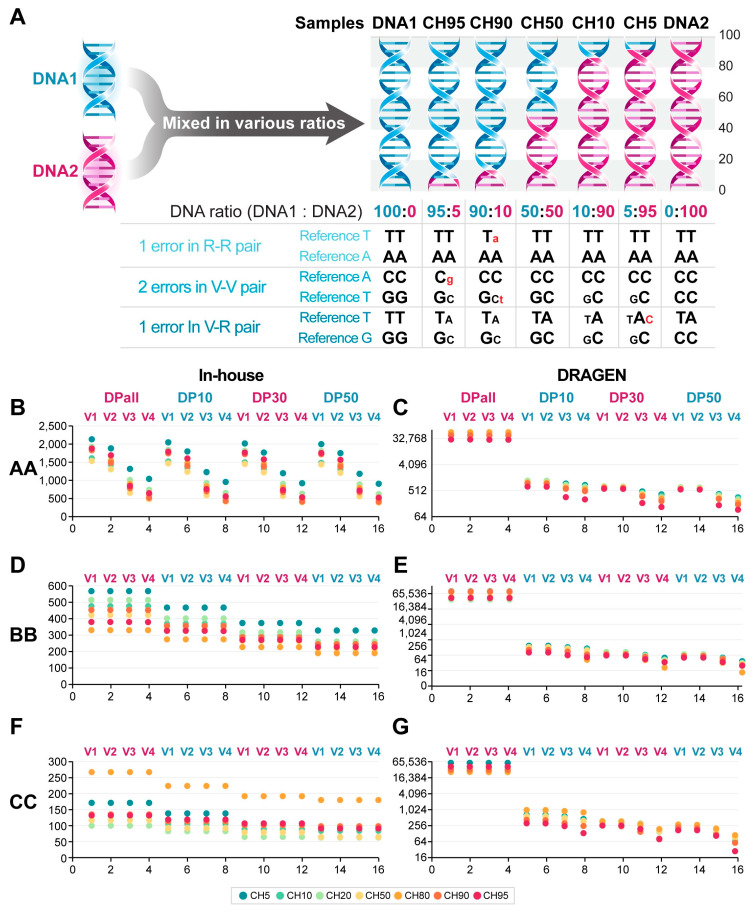
False positive (FP) errors in WES. (**A**). Determination of FP errors based on the pairs of DNA1 and DNA2: R–R pairs, V–V pairs, and R–V pairs. In this example, there is one FP error among 10 R–R pair alleles, two FP errors among 10 V–V pair alleles, and one FP error in R–V pair alleles, resulting in a total FP error rate of 4. FP errors in R–R pair alleles are shown for companies AA, BB, and CC, analyzed by either in-house or DRAGEN methods. (**B**). FP errors in R–R pairs from in-house AA variants. (**C**). FP errors in R–R pairs from DRAGEN-analyzed AA variants. (**D**). FP errors in R–R pairs from in-house BB variants. (**E**). FP errors in R–R pairs from DRAGEN-analyzed BB variants. (**F**). FP errors in R–R pairs from in-house CC variants. (**G**). FP errors in R–R pairs from DRAGEN-analyzed CC variants. In panels (**B**–**G**), the *X*-axis represents various cutoff conditions, and the *Y*-axis shows the number of FP errors. Samples were indicated as CH5 to CH95. V1 to V5 represent VAF cutoffs of 0, 0.01, 0.03, and 0.05, respectively. Similarly, DPall to DP50 represent read depth cutoffs of 0, 10, 30, and 50, respectively.

## Data Availability

All WES fastq files used for this study are publicly available in the NCBI Sequence Read Archive (SRA) under the BioProject accession number PRJNA1178265.

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
