# Peer review of "Enhancing Clinical Applications by Evaluation of Sensitivity and Specificity in Whole Exome Sequencing"

_ijms, 2024, doi:10.3390/ijms252413250_

Round 1

Reviewer 1 Report

Comments and Suggestions for Authors

.

In this study, WES was performed by three companies (AA, BB, and CC) using reference standards composed of DNA from Hydatidiform mole and individual blood at various ratios. The analytical performance of the analysis pipeline was evaluated by artificially controlling the proportion of mutations. 

 WES is commonly used in hospital laboratories for diagnosing rare diseases. However, in the context of rare disease diagnosis, the exact proportion of mutations is typically less critical. This is because qualitative assessments are based on the zygosity of the mutation rather than its quantitative representation.

 As a result, the relevance of evaluating the analysis pipeline based on mutation rates in the context of rare disease diagnosis is questionable. Additionally, mosaicism, being an exceedingly rare occurrence among rare diseases, is not addressed in this issue. 

 From an oncogenetic perspective, WES is considered inadequate in terms of cost-effectiveness. In this study, the analytical performance of the pipeline was evaluated by artificially controlling the proportion of mutations. Despite its limitations, WES is widely used in hospital laboratories for diagnosing rare diseases.

Moreover, key metrics such as mean depth and complexity were not included in the supplementary data, which limits the comprehensive assessment of the analysis pipeline.

Reviewer 2 Report

Comments and Suggestions for Authors

This study assessed the sensitivity and specificity of WES results from three service providers, certified by either the College of American Pathologists (CAP) or the Ministry of Food and Drug Safety (Korean FDA).

Here are comments:

Minor:

1. Line 76. reference 22

R: it should be specified that these results are unpublished (from the instructions for authors "Unpublished data" intended for publication in a manuscript that is either planned, "in preparation" or "submitted" but not yet accepted, should be cited in the text and a reference should be added in the References section). Also considering it has not been yet published, you should not reference it for the methodology description, rather describe the detailed methodology in this paper.

2. DRAGEN – the abbreviation should be introduced in the abstract and the main text the first time it occurs (you introduced it in line 608) Is it DRAGEN or Dragen? Either way, it should be uniform. Also, a brief explanation would be valuable for those who are not from the field.

3. H. mole (Hydatidiform mole) abbreviation should be introduced in the introduction. Provide a brief explanation of this term for the riders who are not familiar and the relevance of its usage here (this should also be in the introduction).

        Figure 1. Why do you present trimmed and untrimmed versions just for AA? Rectangles blue and red should be more pronounced – use thicker lines.

5.        Line 110. It is worth noting that the company names differ from those mentioned in the previous report

R: I think this should be mentioned earlier in the text

6.        Figure 3. Is too large, consider dividing it into two figures.

7.        Lines 408-413 are more for introduction. Also, you need to cite the reference here.

8.        The present study examines sensitivity issues related to WES [9, 10, 17, 20].

R: And what are those references for? Sensitivity issues?

Rephrased

Major:

9.        Considering that you provided great figures representing your results their explanation in the text could be more concise. Also, you discussed the results in the results section. Generally, the results section should be a lot shorter, without discussion, and the most important results better outlined.

10.   You should expand the introduction to provide a better understanding of your paper

Round 2

Reviewer 1 Report

Comments and Suggestions for Authors

 Accept in present form